# Short-Term Benefits from Manual Therapy as an Adjunct Treatment for Persistent Postural-Perceptual Dizziness Symptoms: A Preliminary Prospective Case Series

**DOI:** 10.3390/jfmk9020082

**Published:** 2024-05-03

**Authors:** Brent A. Harper, Larry Steinbeck

**Affiliations:** 1Department of Physical Therapy, Chapman University, Irvine, CA 92618, USA; 2Department of Physical Therapy, Radford University, Roanoke, VA 24013, USA; 3Advanced Rehabilitation, Jasper, GA 30143, USA

**Keywords:** dizziness handicap inventory, falls, fascia, feedback, sensory, postural control, proprioception, risk assessment, vertigo, vestibular

## Abstract

Persistent dizziness and balance deficits are common, often with unknown etiology. Persistent Postural-Perceptual Dizziness (3PD) is a relatively new diagnosis with symptoms that may include dizziness, unsteadiness, or non-vertiginous dizziness and be persistent the majority of time over a minimum of 90 days. The purpose of this case series was to investigate short-term outcomes of reducing dizziness symptoms using a manual therapy intervention focused on restoring mobility in the fascia using a pragmatically applied biomechanical approach, the Fascial Manipulation^®^ method (FM^®^), in patients with 3PD. The preliminary prospective case series consisted of twelve (*n* = 12) patients with persistent complaints of dizziness who received systematic application of manual therapy to improve fascial mobility after previously receiving vestibular rehabilitation. The manual therapy consisted of strategic assessment and palpation based on the model proposed in the FM^®^ Stecco Method. This model utilizes tangential oscillations directed toward the deep fascia at strategic points. Six males (*n* = 6) and females (*n* = 6) were included with a mean age of 68.3 ± 19.3 years. The average number of interventions was 4.5 ± 0.5. Nonparametric paired sample *t*-tests were performed. Significant improvements were observed toward the resolution of symptoms and improved outcomes. The metrics included the Dizziness Handicap Inventory and static and dynamic balance measures. The Dizziness Handicap Inventory scores decreased (i.e., improved) by 43.6 points (*z* = −3.1 and *p* = 0.002). The timed up and go scores decreased (i.e., improved) by 3.2 s (*z* = −2.8 and *p* = 0.005). The tandem left increased (i.e., improved) by 8.7 s (*z* = 2.8 and *p* = 0.005) and the tandem right increased (i.e., improved) by 7.5 s (*z* = 2.8 and *p* = 0.005). Four to five manual therapy treatment sessions appear to be effective for short-term improvements in dizziness complaints and balance in those with 3PD. These results should be interpreted with caution as future research using rigorous methods and a control group must be conducted.

## 1. Introduction

Vestibular disorders can significantly impact an individual’s ability to adequately function in normal daily life activities. It is estimated that 35% (approximately 69 million) of the United States adult population over the age of 40 have experienced some type of dizziness or balance disorder in their lifetime [1]. The latest definition of dizziness by the Bárány Society is “the sensation of disturbed or impaired spatial orientation without a false or distorted sense of motion” [2]. Many vestibular disorders have precise and consistent diagnoses, such as benign paroxysmal peripheral vertigo (BPPV), vestibular neuritis, and Meniere’s disease. A subset of individuals has complaints consistent with a vestibular dysfunction that does not fit neatly into a specific category. There has been an increasing number of articles in the past five to ten years describing “Persistent Postural-Perceptual Dizziness”, also known as PPPD [3,4,5]. 

The etiology and pathophysiology of PPPD, referred to by some as 3PD, is not fully understood, but it does classify as a chronic vestibular disorder [4,6]. It is thought that 3PD is secondary to disruption between or within postural control and visual mechanisms [4,5] and commonly presents after a vestibular disruption, like vestibular neuritis, Ménière’s disease, BPPV, or other medical conditions [5]. Persistent Postural-Perceptual Dizziness symptoms may include dizziness, unsteadiness, or non-vertiginous dizziness and be persistent the majority of time over a minimum of 90 days. Symptoms are provoked by standing or walking in visually complex environments and changing positions from lying to sitting or sitting to standing [6,7,8]. Persistent Postural-Perceptual Dizziness is the most common vestibular disorder for people 30 to 50 years of age [5,9], affects females more than males [10], and is associated with migraines and various psychological conditions, including anxiety and depression [4,5,8].

In 2017, PPPD, or 3PD, was defined by the International Classification of Vestibular Disorders (ICVD) and will be included in the upcoming ICD-11 [3,6] (Appendix B). As a dynamic condition, 3PD symptoms present structurally, functionally, and psychologically, and clinical presentations vary with an individual’s symptom tending to cluster in one of those three influencing areas [6,8]. Persistent Postural-Perceptual Dizziness is a diagnosis of exclusion. Therefore, it is helpful to identify the dominant influencer and to target intervention options toward that area when developing a 3PD plan of care.

Prior to inclusion as a diagnostic category, epidemiological studies reported that up to 25% of patients with complaints of balance disorder did not fall into a specific diagnostic category [3,11]. However, when 3PD is added to the diagnostic criteria, this will be reduced to 15–20% [3,12]. The criteria for inclusion ranges from “symptoms occur without specific provocation” to “symptoms are not accounted for by another disease or disorder” [3]. Studies have shown that ≥25% of acute (i.e., cerebral vascular incident, traumatic brain injury, and deconditioning) and episodic syndromes (i.e., BPPV and Ménière’s) can be precursors to 3PD [3,6,11]. 

In a given year, 53% of reported falls occur in older, chronically dizzy individuals [13]. Thus, 3PD in older individuals might increase the risk of falls. BPPV reoccurrence risk ranges from 26% [14] to 50% [15] and is 1.7-fold higher in older patients than in younger [16]. The increase in falls related to dizziness and 3PD symptoms may be due to the associated functional gait abnormalities, including a slower pace or a more cautious gait pattern as if walking on a slippery surface. Furthermore, those with 3PD demonstrate increased body sway and amplified movement compensations during static or dynamic balance tests [6,8]. 

Easy to perform clinical functional gait assessments (e.g., timed up and go) and balance tests (e.g., tandem and single limb stance) may assist the clinician to gather differential information to determine if persistent dizziness and unsteady feelings are consistent with a 3PD diagnosis and persistent vestibular-like symptoms [6]. If so, the current treatment for 3PD consists of a combination of multimodal interventions, including medication, such as selective serotonin reuptake inhibiters (SSRIs); vestibular rehab/habituation exercises; and cognitive behavioral therapies [6,8,17]. 

Our brains are generally efficient at filtering environmental input, allowing for safe movement through space. Movement efficiency occurs when an individual is transitioning from lying to sitting or standing or during directional mobility, such as crawling, walking, and running. The vestibular system is critically involved in integrating sensory signals from personal space (e.g., somatosensory, proprioceptive, visual, and auditory) and extra-personal space (e.g., visual and auditory). Such inputs become codified in the brain through the central vestibular system afferents and peripheral proprioceptive afferents [18]. Vestibular, visual, and proprioceptive cues influence postural corrections in the normal adult. However, normal aging affects proprioceptive inputs and therefore older individuals may become more sensitive to further distortions in proprioceptive input, negatively affecting balance and motor control [19].

Multiple studies [20,21,22] in the past two decades have brought to light the capsulated (Pacini, Ruffini, and Golgi) and unencapsulated (free nerve) endings of mechanoreceptors and proprioceptors located within the deep muscular fascia in the limbs, trunk, head, and face regions. It is possible that alterations within the deep fascia (through its interaction with the peripheral proprioceptive elements and encapsulated and free nerve endings) can be involved in impaired proprioception and may influence 3PD symptoms. 

It is hypothesized [21] that restoring the mobility of the deep fascia may normalize proprioception information, decreasing symptoms and improving function. Manual frictions using tangential oscillations are directed toward the muscle and deep fascia at strategic points based on the model presented in the Fascial Manipulation^®^ Stecco Method. The goal of the manual manipulation is to reduce or resolve local tenderness and normalize slide glide with deep fascia in regard to the underlying muscle. See Appendix C for an overview. If this is the case, then adding a manual therapy intervention to a 3PD treatment plan may provide another intervention option to manage symptoms and improve quality of life. The current treatments already improve quality of life, allowing individuals to return to normal function within a shorter time frame [17]. Adding manual therapy to the current multimodal treatment option may result in even faster progress. The purpose of this case series was to investigate short-term outcomes of a manual therapy intervention focused on restoring mobility in the fascia using a pragmatically applied biomechanical approach, the Fascial Manipulation^®^ (FM^®^), in subjects with 3PD. 

## 2. Case Presentations

### 2.1. Case Series Study Design 

A preliminary prospective case series and analysis of pre-existing de-identified data were conducted to identify patients who continued to complain of dizziness symptoms despite previously receiving vestibular rehabilitation, including the treatment of any cervicogenic pain generators, and who received manual therapy after completing these other treatment interventions. The cases were identified and extracted between June 2019 and December 2021. Twelve cases of individuals who had received manual therapy intervention due to persistent dizziness and balance symptoms following previous interventions addressing the vestibular and cervicogenic regions and who fit the prospective 3PD diagnosis were identified in 2017 by the ICVD (Appendix B), at minimum, 3 to 6 months after other vestibular disorders were excluded by the medical referral source. All the participants signed an agreement to consent to treatment prior to data collection and were informed that their de-identified records and follow-up data outcome information may be used for research purposes upon discharge. This study was conducted according to the guidelines of the Declaration of Helsinki and approved by the Institutional Review Board (or Ethics Committee) from Radford University. The treating clinician performing the manual therapy interventions was a licensed physical therapist with over 30 years of experience. The same individual extracted and synthesized the outcome data but was blinded to statistical analysis.

### 2.2. Subjects: Cases

A total of 12 patients were included. The patients were initially referred to an outpatient physical therapy clinic for non-central and non-peripheral-related unspecified dizziness, which was consistent with a 3PD-type presentation (Appendix B). According to the referral sources, the individuals received medical work up and were negative for BPPV, vestibular neuritis, Ménière’s, vestibular migraine, vestibular schwannoma, vascular insult, or any other brain mass. In addition, these individuals had not experienced acute or chronic neck pain within the last three months, nor did they present with any cervical range of motion deficits at the time of this manual therapy procedure. Associated neck pain is one criterion associated with the diagnosis of cervicogenic dizziness [23,24]. Furthermore, specific upper cervical spinal segmental mobility (e.g., C0–C1, C1–C2, and C2–C3) limitations had been treated previously or were not limiting variables at the time of the soft tissue manual therapy intervention. Other prior treatments may have included canalith-repositioning maneuvers and vestibular rehabilitation with the resolution of vertiginous episodes; however, recalcitrant symptoms of dizziness persisted as the patients had plateaued with prior standard care. It should be noted that current common interventions for 3PD include patient education about the condition, vestibular rehabilitation (i.e., canalith-repositioning and habituation exercise), medication (i.e., selective serotonin reuptake inhibitors (SSRIs), and cognitive behavior therapy [8]. When referred for this soft tissue intervention, the participants had completed prior vestibular rehabilitation therapy and had failed to progress further, plateauing, with continued symptoms. Referral occurred, therefore, around 3 to 6 months after completing the original services. The only addition to the participants’ treatment plan was the fascial soft tissue intervention (FM^®^).

### 2.3. Examination: Outcome Measures 

At two time points before and after intervention (pre- and post-test), each participant completed the Dizziness Handicap Inventory (DHI) and three commonly utilized clinical performance measures, the timed up and go (TUG), single leg stance (SLS), and tandem stance. Baseline data were collected prior to the initiation of FM^®^ (pre) and during the final treatment visit (post-testing). The DHI measures subjective complaints commonly experienced by those with 3PD, while the balance measures were assessed because over a quarter of those with 3PD tend to have postural stability deficits [3,11] where disruptions in balance may lead to an increased risk of falls. Because BPPV tends to be a precursor to 3PD [3,6,11], the cases included had previously had vestibular rehabilitation to address the original symptoms. 

The DHI is a subjective patient-reported outcome (PRO) measure consisting of a 25-item questionnaire quantifying the impact of dizziness on daily life by measuring the self-perceived handicap using a score between 0 and 100, with a higher score indicating a greater perceived handicap due to dizziness [25]. The DHI has an excellent reliability for the total score (*r* = 0.97 and *p* < 0.001) and internal consistency (alpha 0.89) with a minimal detectable change (MDC) of 17.18 [26]. 

The timed up and go (TUG) is a functional test traditionally used as a fall risk assessment. The TUG was performed beginning with the participant sitting in a chair. The participant was instructed to stand up when they heard the word “Go”, walk 3 m (9.8 feet) at a comfortable pace, turn around, walk back, and sit down. The entire sequence was timed in seconds beginning with when the participant rose to stand and ending when the participant sat again. Each participant had one or two practice trials prior to baseline data collection to become familiar with the task. The TUG was selected as the functional mobility assessment metric because it has excellent reliability (ICC = 0.97) [27] for older adults living independently in the community, including those with a variety of medical conditions (ICC = 0.99) [28], and an excellent correlation with gait speed (*r* = 0.66) [29].

The single limb stance (SLS) was performed on each leg. The participant stood, with eyes open, on one leg, with the non-stance hip flexed to approximately 30° and the knee flexed to approximately 45° and with hands on hips, for as long as possible. The test was stopped if the participant touched the ground with their foot or if their hands left their hips. The SLS was selected as a simple clinical metric because it has excellent reliability (ICC = 0.85 to 0.99) [30,31]. Furthermore, a recent systematic review [32] identified the SLS to be the most useful non-instrumented balance test to predict falls and to differentiate between fallers and non-fallers. An increased SLS time is a potential indicator of decreased fall risk.

During the third functional test, the tandem stance, the participant stood with their feet on a straight line, placing either the right or left foot in front of the other. In this study, data were collected on both sides, with the right foot in the front position and with the left foot in the front position, with the hands placed on hips with eyes open. The test was timed in seconds and measured the length of time the participant could maintain balance. It was stopped if the participant lost their balance or moved their hands or feet from the starting position. The test was limited to 30 s to account for a ceiling effect from the test [33]. A systematic review [32] identified that the tandem stance was a useful, non-instrumented postural control test; however, it has been used less frequently in clinical studies.

### 2.4. Plan of Care: Manual Therapy Intervention

One physical therapist with 30 years of experience completed a physical assessment followed by a pragmatic soft tissue manual therapy intervention based on a biomechanical (FM^®^) method. The assessment consisted of active and passive movements of the neck through the three cardinal planes followed by strategic palpation of the deep fascia over the centers of coordination (CCs) and centers of fusion (CFs) [34]. Palpation followed the ranking system described by Cotti et al. [35] in which each palpated point was identified based on patient-reported tenderness and therapist-perceived tissue stiffness. The patients were followed one to two times per week until the symptoms were resolved or a maximum of eight visits.

The treatment involved manual manipulation consisting of deep pressure with tangential oscillations (non-gliding manual friction), which are performed in multiple directions based on clinical assessment and the therapist’s perception of densification, or a lack of fascial gliding, and which may also correlate with symptom reproduction or pain (see Figure 1A,B). This soft tissue manipulation continues until the therapist no longer perceives the densification and the patient reports a decrease in symptom provocation or pain intensity by at least 50% based on a numerical pain rating scale (NPRS) [36,37,38]. A full description of the FM^®^ approach is beyond the scope of this manuscript, but previous published manuscripts describe this approach [37,39,40]. See Appendix C for an overview.

### 2.5. Plan of Care: Home Exercise Program

The patients were instructed to continue with any formal home exercise program (HEP) previously provided during vestibular rehabilitation. A prior HEP may have included habituation exercises for vestibular ocular reflexes, canalith-repositioning exercises (even if asymptomatic), lower extremity strengthening (hip, knee, and ankle), lower extremity flexibility (ankle and toes), or standing balance (semi tandem, tandem, and single limb). If the patient had not been performing a home exercise program, one was initiated to resolve any deficits observed with the lower extremity strength or flexibility, classified as general exercise. 

### 2.6. Statistical Analysis

A nonparametric analysis of the Wilcoxon signed-rank test for paired samples was performed for each metric. The Shapiro–Wilk test was used to test for normality of the data. All the analysis was performed using IBM SPSS Statistics Version 27 software (International Business Machines Corp., Armonk, NY, USA).

## 3. Case Presentations: Outcomes before and after Intervention

### 3.1. Subjects: Cases

Twelve subjects met the inclusion criteria during the time period previously identified, having received prior vestibular rehabilitation and other treatments to resolve any competing causative pain generators (e.g., cervicogenic region) prior to receiving the soft tissue manual therapy. The participants were evenly distributed between males (*n* = 6 or 50%) and females (*n* = 6 or 50%). The mean age was 68.3 ± 19.3 years with a minimum of 17 and maximum of 86 years, giving a range of 69 years of age and median of 74.5. The male mean age in years was 76.8 ± 10.7. The mean female age was 59.7 ± 23.1 (Appendix A). There was no significant difference between the various metrics (i.e., variables) and gender.

### 3.2. Number of Visits

The mean number of visits was 4.5 ± 0.5, ranging from a minimum of 2 and a maximum of 8, giving a range of 6 visits and a median of 4, and no adverse events from the FM^®^ were reported. These visits occurred over a period of time ranging from one to four weeks depending on the individual case and number of treatment sessions they received. Thus, short-term changes are defined by the episode of care provided (Appendix A).

### 3.3. Dizziness Handicap Inventory 

A nonparametric paired sample *t*-test was run to determine if there was a significant difference between the Dizziness Handicap Inventory (DHI) scores before and after the FM^®^ treatment. The DHI pre scores (*n* = 12) had a significant Shapiro–Wilk (*p* = 0.043) and the DHI post scores (*n* = 12) had a non-significant Shapiro–Wilk (*p* = 0.132). The Wilcoxon signed-rank test for the paired samples showed a statistically significant result (*z* = −3.1 and *p* = 0.002). The DHI scores decreased (i.e., improved) by 43.6 points (*z* = −3.1 and *p* = 0.002), from a DHI pre of (Mean/SD) 53.8 ± 13 to a DHI post of (Mean/SD) 10.2 ± 5.6 (Figure 2).

### 3.4. Timed up and Go

A nonparametric paired sample *t*-test was run to determine if there was a significant difference between the timed up and go (TUG) scores before and after the FM^®^ treatment. The TUG pre scores (*n* = 12) and TUG post scores (*n* = 12) had a non-significant Shapiro–Wilk (*p* = 0.323 and *p* = 0.573), respectively. The Wilcoxon signed-rank test was significant (*z* = −2.8 and *p* = 0.005), indicating that the gains made for the TUG during the treatment were significant. The TUG scores decreased (i.e., improved) by 3.2 s (*z* = −2.8 and *p* = 0.005) from a pre of (Mean ± SD) 11.7 ± 2.6 to a post of (Mean ± SD) 8.5 ± 2.9 (Figure 2). 

### 3.5. Tandem Stance

A nonparametric paired sample *t*-test was run to determine if there was a significant difference between the tandem stance scores for the right and left test positions before and after the FM^®^ treatment. The tandem pre left scores (*n* = 12) and tandem post left scores (*n* = 12) had a non-significant Shapiro–Wilk (*p* = 0.149 and *p* = 0.431), respectively. The Wilcoxon signed-rank test was significant (*z* = 2.8 and *p* = 0.005), indicating that the gains made for the tandem left during treatment were significant. The tandem pre right scores (*n* = 12) had a significant Shapiro–Wilk (*p* = 0.017) and the tandem post right scores (*n* = 12) had a non-significant Shapiro–Wilk (*p* = 0.091). The Wilcoxon signed-rank test was significant (*z* = 2.8 and *p* = 0.005), indicating that the gains made for the tandem right during the treatment were significant. The tandem left increased (i.e., improved) by 8.7 s (*z* = 2.8 and *p* = 0.005) with pre scores (Mean ± SD) 8.2 ± 6.9 and post scores (Mean ± SD) 16.8 ± 8.4. The tandem right increased (i.e., improved) by 7.5 s (*z* = 2.8 and *p* = 0.005) with pre scores (Mean ± SD) 8.0 ± 8.2 and post scores (Mean ± SD) 15.5 ± 8.1 (Figure 3).

### 3.6. Single Limb Stance

A nonparametric paired sample *t*-test was run to determine if there was a significant difference between the SLS scores for the right and left test positions before and after the FM^®^ treatment. The SLS pre left scores (*n* = 12) had a significant Shapiro–Wilk (*p* = 0.035). For the SLS post left scores (*n* = 12), normality was met with a non-significant Shapiro–Wilk (*p* = 0.379). The Wilcoxon signed-rank test was significant (*z* = 2.9 and *p* = 0.003), indicating that the gains made for the SLS left during the treatment were significant. The SLS pre right scores (*n* = 12) had a significant Shapiro–Wilk (*p* = 0.026) and the tandem post right scores (*n* = 12) had a non-significant Shapiro–Wilk (*p* = 0.267). The Wilcoxon signed-rank test was significant (*z* = 2.9 and *p* = 0.004), indicating that the gains made for the SLS right during the treatment were significant. The SLS left increased (i.e., improved) by 6.8 s (*z* = 2.9 and *p* = 0.003) with pre scores (Mean ± SD) 6.9 ± 7.4 and post scores (Mean ± SD) 13.7 ± 9.5. The SLS right increased (i.e., improved) by 6.6 s (*z* = 2.9 and *p* = 0.004) with pre scores (Mean ± SD) 7.4 ± 7.9 and post scores (Mean ± SD) 14.0 ± 9.0 (Figure 3).

## 4. Discussion

### 4.1. The Effect of Manual Therapy on Each Metric in This Case Series

The dramatic significant improvement in the DHI scores post-FM^®^ (Figure 2) met the MDC of 17.18 [26]. Furthermore, the participant DHI scores improved from pre scores of 53.8 ± 13, indicating high-end “Moderate” dysfunction, to post-test scores of 10.2 ± 5.8, signifying lower-end “Mild” dysfunction [41]. 

Prior research identified patients with TUG scores ≥ 13 s as being at increased risk of falling [42,43]. Although none of the participants in this study scored high enough to qualify as at risk for falls during the initial evaluation, their scores still demonstrated statistical improvement after receiving FM^®^. According to Kear et al. [44], a normative TUG reference value for people aged 20–59 is 8.9 s with an average range of 6.0 to 14.5 s. The TUG scores (Figure 2) after the FM^®^ intervention decreased 3.2 s overall and the post TUG scores appeared to improve, decreasing into the range of normal values (8.5 ± 2.9).

Although meaningful cut scores for the tandem stance are not widely accepted, Hile et al. [33] classified tandem stance hold time performance for those who could initially stabilize without support. Those holding the tandem stance for <10 s were considered “Low” performers, while those between 10 and 29 s demonstrated “Medium” performance. Those holding the maximum hold time of 30 s were considered “High” in performance [33]. The significant improvements in the tandem stance with both the right and left foot placed in the forward position demonstrated that the participants, as a whole, moved from “Low” to “Medium” performance; however, it is unknown if these are clinically meaningful changes (Figure 3). 

Despite significant gains in the SLS left (6.8 s) and SLS right (6.6 s), it is unclear whether these changes have clinical significance or are enough to assess detectable change. There are different MDC cut scores depending on the patient population. The MDC is the minimal amount of change in scores that must be reached in order to reflect a true or valid difference in scores, not due to chance. An MDC of MDC_90_ or MDC_95_ would indicate that a true change will occur 90% or 95% of the time. In order to be meaningful, the MDC_95_ for the SLS in older adults is 24.1 s [45]. Research using the SLS on various conditions has identified an MDC_90_ of 4.1 s for those with Alzheimer disease [46], an MDC_95_ of 2.7 s for older adults with COPD [47], and an MDC_95_ of ≥9 s for those diagnosed with multiple sclerosis [48]. However, those with 3PD have not had an MDC developed for the SLS. What is known is that significant changes were achieved in SLS post manual therapy treatment in this case series. What is not known is if this change is or is not clinically meaningful (Figure 3).

In general, the patients were pleased with the gains made from the FM^®^ manual therapy intervention as they improved in overall function based on the subjective and objective metrics collected in this study. These subjective reports appear to correlate with the statistically significant improvements in the subjective DHI scores and the objective metrics scores from the TUG. 

### 4.2. Importance of Adding Non-Pharmacological Interventions for 3PD

The results of this case series provide initial findings regarding the assessment and treatment aimed at restoring fascial mobility, which might be a helpful non-pharmacological short-term treatment option in patients with 3PD, adding to the current multimodal treatment selection. This case series showed significant improvement in patient-reported outcomes (e.g., DHI) and several common clinically objective tests and measure metrics (e.g., TUG, tandem stance, and SLS). Because this is a case series, the findings are not generalizable, nor can observed changes directly relate to the intervention because there was no control group. Despite this limitation, the results identified from this case series indicate that FM^®^ may be a beneficial short-term non-pharmacological adjunct treatment for those with 3PD and worth investigating with a larger, homogeneous, pilot study or randomized clinical trial study with a control group. Because 3PD may significantly impact an individual’s quality of life and function, it is important to identify additional avenues of intervention.

The pathophysiology of 3PD remains unclear [49,50]. Theoretical approaches focus on visual, behavioral, and vestibular integration errors, which produce a reduced cortical integration of spatial cues in response to a triggering event. In general, symptoms associated with 3PD are worse in the upright position, exacerbated by changing visual stimuli, and may be impacted by active or passive head motions. People may complain of vague symptoms, such as a full sensation in their head, haze, or cloudiness. During ambulation, they may feel unsteadiness or veer off course. It often takes simple contact to balance or stabilize themselves prior to movement [3,8,10]. Therefore, non-pharmacological manual therapy interventions that may improve static or dynamic balance to enhance physical mobility might be valuable in further improving symptoms in those with 3PD.

### 4.3. Value of Clinical Balance Assessments

A recent systematic review reported that center of pressure (CoP) tests might be more sensitive when predicting falls than velocity measurements [32]. Although the current study did not utilize force plates to identify CoP or sway velocities, there were increased time measurements with standard clinical tests related to CoP testing, and the gains made in this study may be an indicator of decreased fall risk and reduction in dizziness symptoms. Therefore, improving balance stance times (e.g., tandem and SLS) may be a valued clinical indicator of patient improvement.

### 4.4. Prior Research Supporting the Soft Tissue Intervention

The FM^®^ method has demonstrated favorable clinical findings when applied to other conditions involving patellar tendinopathy [51], chronic ankle sprains [52], whiplash [53], and chronic shoulder pain [39]. Hypothetically, normalizing fascial mobility by addressing specific restricted fascial points using the method might have been one causative aspect involved in the subjective and objective gains in this case series. Postural control and perception of the body’s orientation and objects’ location in extra-personal space require the integration of proprioceptive, visual, and vestibular signals [18,54]. Other studies have shown that the restoration of fascial mobility using the FM^®^ method has resulted in a significant gain in those with low back pain [40], and a recent systematic review concluded FM^®^ decreased pain and disability [55]. Furthermore, restoring the fascia’s ability to glide and slide using the FM^®^ method appears to result in improved integration of the postural control system to improve balance [56,57] and enhance motor performance, including reaction time [58]. 

FM^®^, as an added treatment to the multimodal approach, may have been integral to the improvement in DHI patient-reported outcome scores and static and dynamic performance (e.g., tandem stance, SLS, and TUG) for those who had plateaued after standard care by normalizing fascial mobility, which may play a role in improving the integration of postural control. Currently, no other known studies have measured the potential effects of FM^®^ on those with persistent dizziness or 3PD.

### 4.5. How Changes Might Have Occurred after FM^®^

Foster [59] points out that the proprioceptors in the neck are well positioned to amplify sensory information through an abundance of mechanoreceptors in the deep, segmental neck muscles. Brandt and Bronstein [60] support this idea by pointing out that unilateral electrical stimulation to the cervical musculature can produce an illusion of movement. Postural control and perception of the body’s orientation and objects’ location in extra-personal space require the integration of proprioceptive, visual, and vestibular signals [18,54]. Brandt [61] notes that proprioceptive input from the neck coordinates with feedback from the eyes, head, and body posture, as well as spatial orientation. Unsteadiness could be related to a disorder in the vestibular, visual, vascular, or cervicoproprioceptive mechanisms. Fascial Manipulation expands this concept by examining the proprioceptive structures in the trunk and the limbs in relation to the neck and head. The techniques chosen addressed the fascial mobility limitation in the affected region. Therefore, restoring fascial mobility in the head and neck may normalize peripheral input, decrease sensory conflict, improve postural control, and decrease symptoms.

Proprioceptive afferents have been described as organized in a metameric distribution similar to dermatomes, myotomes, and sclerotomes [21,62], and these afferents could be a source of radiating symptoms [63]. This radiation of symptoms in the skull may be likened to a paresthesia or dysthesia in the limbs. Individuals have a difficult time precisely describing the location and description of symptoms. In addition, Stecco [64] expands on the concepts of Brandt in correlating the role of the gamma efferent/muscle spindle relationship to dizziness by including the relationship of the limbs, head, neck, and trunk through these proprioceptive elements. If this is correct, it may provide another explanation for the therapeutic effects gained by addressing the mobility of the fascial system.

It has been common in medicine to discuss the impact of manual or mechanical interventions on the fibrous component of fascia. However, there has been a growing interest in the extracellular matrix (ECM) role regarding interventions. The role of the ECM in pain control has been poorly understood. However, there has been a rapid increase in research being presented on the role of intercellular contents, particularly glycosaminoglycans and their influence on myofascial pain [65,66]. Variations in the molecular weight of hyaluronan influence nociception and inflammation. Low-molecular-weight hyaluronan (LMHA), 250 kDa, has been shown to reduce joint elasticity and the threshold of local nociception via influence at cell-binding cite CD-44 [67,68,69]. Due to the sensitivity to stressors, HA has been observed to aggregate on itself, hence altering the viscosity in the surrounding tissue. Improvement in motor function has been reported on post-stroke spastic limbs following injection of the enzyme hyaluronidase, which impacts the molecular weight of HA by initiating fragmentation of the molecular chains leading to restoration of the homeostasis of the tissue viscosity in the ECM [70,71]. Modeling has shown that the ECM, particularly HA, is malleable and plastic under stresses and strains created with manual manipulation, including FM^®^ intervention [72,73]. Manual therapy, via deep friction (tangential oscillations), may influence the self-aggregated chains to catabolize or fragment them, resulting in a cascade of decreasing molecular lengths resulting initially in local inflammation and ultimate restoration to normal physiologic connective tissue properties. With the restoration of fascial gliding, the brain receives accurate proprioception and length tension input allowing for optimal motor unit recruitment (Appendix C). This discussion point is essential as the restoration or homeostasis of HA is a crucial factor that, when restored, results in the normalization of fascial mobility leading to positive changes in subjective and objective clinical metrics [37,64,72]. 

### 4.6. Limiting other Potential Pain Generators for Continued Symptoms

The diagnosis of “cervicogenic dizziness” presents with similar symptoms to those described as 3PD; however, they have a different clinical presentation. Much has been published regarding diagnoses associated with mechanical problems involving the cervical spine, which was initially described as cervicogenic vertigo. It has evolved to be known as cervicogenic dizziness to reflect the differences between the term’s vertigo and dizziness [23,24,74]. Symptoms described with cervical dysfunctions can mirror symptoms associated with 3PD, particularly anxiety, fear avoidance, catastrophizing, kinesiophobia, and depression [75,76]. Breinbauer [77] utilized a test to assess spatial navigation and found a significant difference in the altered navigational network associated with patients diagnosed with 3PD versus non-3PD diagnoses. This may account for some of the similar subjective descriptions between patients with persistent pain and 3PD. Despite similarities, 3PD is unique, and restoring fascial mobility may be one way to further improve this condition.

### 4.7. Limitations

The primary limitation of this study is that it is a case series that lacks a control group, not allowing direct cause and effect from the intervention. Persistent Postural-Perceptual Dizziness is considered a type of disorder of the nervous system resulting in dizziness and secondary functional gait and balance issues that can be correlated with significant cognitive-affective predisposing factors, such as anxiety and depression. Although the DHI composite score comprises three domains (e.g., functional, physical, and emotional), the total score was the only information utilized for this study. Evaluating the DHI emotional score separately might have provided useful information explaining improvements. However, information regarding participants susceptible or with anxiety and depression was not collected. Furthermore, prior or current interventions the participants might have received, such as for emotional health, were not collected. Those with 3PD include a wide age distribution, suggesting 3PD may not be age-dependent. This was reflected in our sample, which included individuals aged 17–86 years. However, it should be noted that older individuals tend to have poorer standing balance on one leg (SLS) or two legs with a narrow base of support (tandem). Although the SLS and tandem improved statistically, they did not improve drastically. This may be due to the fact that 75% of the participants were over 71 years of age and may have had age-related balance deficits. The only addition to the participants’ treatment plan was the fascial soft tissue intervention (FM^®^). Despite this limitation, this case series supports the need to perform a larger and controlled pilot study or randomized clinical trial to assess the benefit of adding this non-pharmacological intervention to those with signs and symptoms consistent with 3PD. 

## 5. Conclusions

This case series study was conducted in preparation for a future pilot study or randomized clinical trial. This case series presents preliminary short-term findings that adding a pragmatically applied biomechanical manual therapy intervention (e.g., FM^®^) to the multimodal plan of care may enhance function and subjective outcomes for patients with residual dizziness after receiving vestibular rehabilitation, without any apparent cervicogenic pain generators, which may be classified as having 3PD, often a diagnosis of exclusion. Research has attributed an increasing role of the deep muscular fascia in pathology, motor control, proprioception, and perception in recent years. Clinical improvements may be directly related to FM^®^ intervention independently or in conjunction with a vestibular rehabilitation program or other unknown variables. However, these results should be interpreted with caution as future research using rigorous methods and a control group is needed. The pragmatic biomechanical manual therapy approach described in this case series may suggest an avenue for future research and the potential development of effective alternative manual therapy interventions for 3PD to determine if the FM^®^ method, when added to current treatments, more quickly decreases dizziness, improves postural control, and enhances quality of life in those with persistent dizziness or 3PD symptoms.

## Figures and Tables

**Figure 1 jfmk-09-00082-f001:**
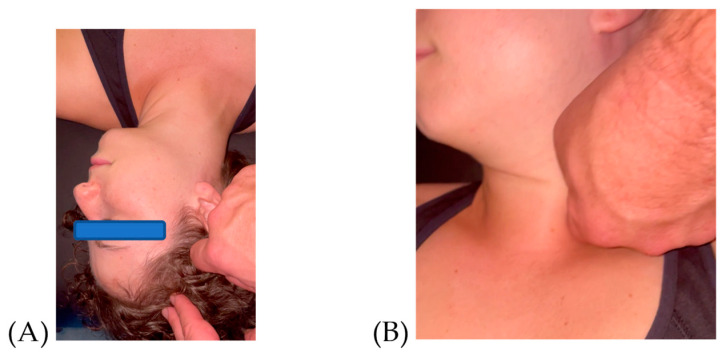
(**A**) Lateral Motion–Caput 2 (LA-CP2), mid portion of anterior temporalis; (**B**) Intrarotation–Collum (IR-CL), between sternal and clavicular portion of the sternocleidomastoid muscle, over the omohyoid.

**Figure 2 jfmk-09-00082-f002:**
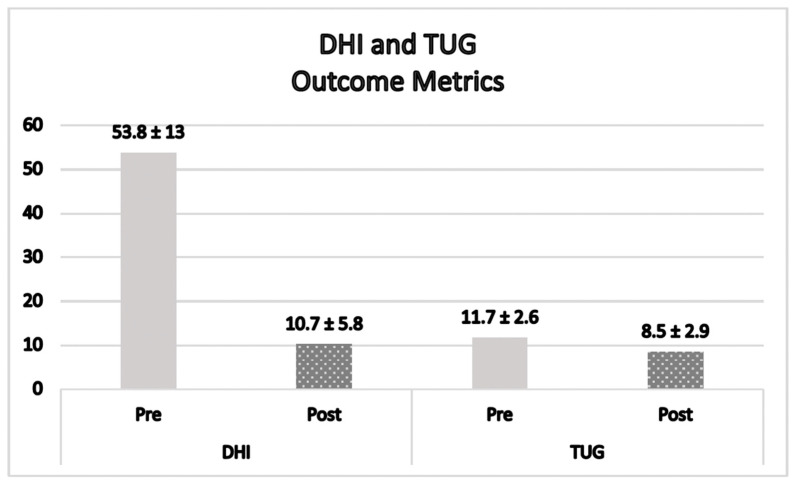
DHI = Dizziness Handicap Inventory, lower score is better, in points; TUG = timed up and go, lower score is better in seconds; before intervention is pre = pre-test score; after intervention is post = post-test score.

**Figure 3 jfmk-09-00082-f003:**
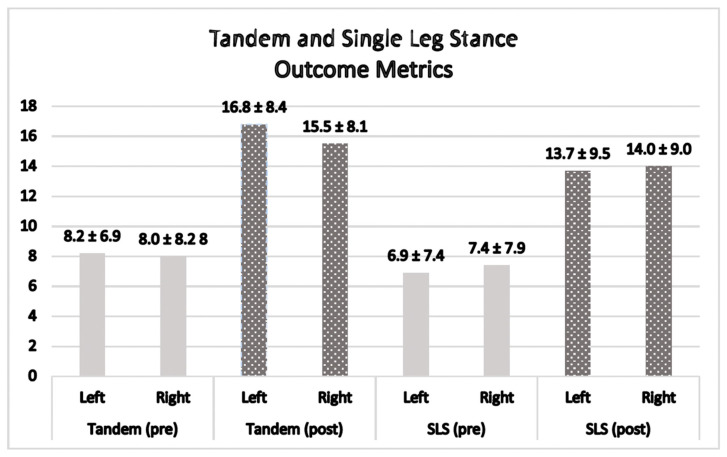
Tandem stance, higher score is better, in seconds; SLS = single limb stance, higher score is better, in seconds; before intervention is pre = pre-test score; after intervention is post = post-test score.

## Data Availability

The datasets used and/or analyzed during the current study are available from the corresponding author upon reasonable request.

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
