# Peer review of "Short-Term Benefits from Manual Therapy as an Adjunct Treatment for Persistent Postural-Perceptual Dizziness Symptoms: A Preliminary Prospective Case Series"

_jfmk, 2024, doi:10.3390/jfmk9020082_

Round 1
Reviewer 1 Report (New Reviewer)
Comments and Suggestions for Authors
1) The term. Unify the terms. PPPD and 3PD.
2) What is PPPD? I cannot understand the pathophysiology of it. Explain it clearly.
3) Table1 and Fig. 2 are the same things. Delete table 1.
4) Table 2 and Fig. 3 are the same things. Delete table 2.
5) I cannot understand the mechanism of treatment. Figure 1 does not show the detail of therapy. You should add some illustrations which show the principle of treatment.
6) To elucidate the efficacy of treatment, comparison to the control group is needed. Add the data of control group, because most of vestibular disease cure spontaneously.
7) Examinations are weak. Are they quantitative balance tests? What is TUG? TUG is not a common test. Add a movie or illustration. You should show the data of healthy human. What is SLS? You should add a movie or illustration. You should show the data of healthy human. Tandem stance. You need to add a data of healthy human.
8) Why was the treatment effective? You should consider and describe the mechanism.
9) Figure. You should describe the unit in a graph.
Comments on the Quality of English Language
quality of english is low.
Author Response
Please refer to the attached file.

Reviewer 2 Report (New Reviewer)
Comments and Suggestions for Authors
This manuscript presented research findings on auxiliary treatment methods for short-term recovery of the vestibular system. It systematically analyzed the effectiveness of the proposed methods.
This manuscript needs to be revised and supplemented in the following contens.
Q1) Please include the name of a university that acquired the approval of the institutional review board in line 135.
Q2) In the statistical analysis, the Wilcoxon signed-rank test, a nonparametric analysis, is different from a paired sample t-test, a parametric analysis. Please specify the statistical analysis method that was used correctly in the study. From sections 3.3 to 3.6, the Shapiro-Wilk test was also used to test the normality of data. Please include this method in section 2.6 as well. The statistical analysis software also needs to be indicated in the study.
Q3) Please elaborate on the meaning of MDC95 and MDC90 in section 4.1.
Q4) Please check whether the contents of Sections 4.5 and 4.6 are related to the scope of this study.
Q5) "Section 4.6. Limitations" should be revised to "Section 4.7. Limitations."
Q6) Please correct the misspellings such as mediation (line 156) and indictor (line 197).
Round 2
Reviewer 1 Report (New Reviewer)
Comments and Suggestions for Authors
Study design is too weak. “However, PPPD (3PD) pathophysiology is not well understood by the medical community at present.”
It is impossible to treat a vague disease. This study does not have scientific finding.
Comments on the Quality of English Language
quality od english is low.
This manuscript is a resubmission of an earlier submission. The following is a list of the peer review reports and author responses from that submission.
Round 1
Reviewer 1 Report
Comments and Suggestions for Authors
Congratulations for the exhaustive description of the case report and for the originality of the topic.
Please include a brief description in the abstract of the manual therapy techniques used in the treatment protocol.
Also use the abbreviations for Persistent Postural-Perceptual Dizziness in the abstract also.
Some information from the conclusions section of the abstract may not be relevant. Please shorten it.
Please include some more information on the manual therapy techniques in the introduction section.
Please discuss a little bit more on neurophysiological mechanisms of fascial techniques in the discussion section.
Author Response
Thank you for your comments and suggestions. We have responded to each item. The attached file has our responses followed by the manuscript with changes, and each Appendix to support your review. Thank you.

Reviewer 2 Report
Comments and Suggestions for Authors
This is an uncontrolled yet statistically analyzed small case series on manual therapy in patients with 3PD. Since data on body-oriented treatment in all kinds of functional disorders are needed, especially reports from physiotherapy, movement therapy, occupational therapy and psychotherapy, and especially if they promote integrative non-pharmacological approaches, I suggest to publish it, but only after a profound revision and with much more differentiated discussion (especially the problem of FM as a passive intervention, and of the many methodical shortcomings).
The authors should discuss 3PD not only as a chronic vestibular and dynamic but also as a functional neurological disorder, with disturbances in central nervous top-down regulation, including a considerable cognitive-affective component (see, for instance, the cited works by Popkirov, Stone and others). I recommend to read (and maybe cite) https://neurosymptoms.org/en/symptoms/fnd-symptoms/functional-dizziness-pppd/. To mention this in the limitations (“3PD can be considered a type of functional neurological disorder which is correlated with predisposing factors such as anxiety and depression”) appears too uninformed and unconnected.
I think it is important to draw an as-good-as-possible line to BPPV and dizziness due to frailty and multimorbidity in old age, but this is quite blurred in lines 74ff. Considering this, the age distribution in this sample is suboptimal (imagine an SLS at age 86!) – this should be discussed as an important limitation.
Timelines are needed. How long took the rehab before FM? How much time passed in between? How long took the HEP? Please specify the two time points before and especially after intervention: Directly after the FM or the HPE? Days, weeks or months later? The preceding rehab, the time between rehab and manual therapy in study participants (and ideally the natural course of dizziness of people after rehab but without manual therapy) should be described.
· 2.1.: Apparently, participants have had manual therapy before (line 124f)? The authors should explain the difference between the previous and the new manual intervention. Otherwise it is most likely that previous effects had been and current effects will be transient.
· The observed improvements might be attributed to delayed long-term effects of active vestibular rehabilitation that manifested at the time when patients returned to their lives.
· Apparently, participants have had vestibular rehabilitation addressing the vestibular and cervicogenic regions before. It remains unclear, why the FM was performed after rehab, and not (as usual) as part of a multimodal rehabilitation program.
· Usually, rehab is a predominantly active approach, and manual therapy is predominantly passive. For various reasons (high placebo effect, increased passivation, decreased self-efficacy and control), the current literature and evidence favors active over passive approaches (to all kinds of functional disorders). This should be discussed.
· Especially since patients continued with “any formal HEP”, i.e. further active exercises, well described as, for instance, exercises for habituation, strengthening, flexibility, and balance. This resembles a typical and effective treatment for functional gait or balance disorders. According to the theory of 3PD being a functional disorder, it is most likely that rather these active than the preceding passive manual therapy explains the improvements, or that the improvements result from delayed effects from the former active rehab preserved by the later active HEP.
This case series would be more interesting if only half of the patients would have been instructed to continue with active exercises, or only half of the patients would have received FM - and both subgroups would have been compared.
The “Fascial Manipulation®” intervention is meant to improve fascial mobility and mechanoreceptor/proprioceptor function in a mostly passive manner. It is a registered trade mark and it hasn’t become transparent to me if the authors might have any conflicts of interest here.
Since this is a small case series, a few examples from specific patients would be interesting. For example, what was the outcome of the patient who had the maximum of 8 interventions? What were the characteristics of the patients that benefitted the most? In this respect, I suggest to include Appendix C in the manuscript.
Since all three domains of the DHI carry important information, the results of the subscales should be added. If the total score was calculated, their description (and analysis) should be easy.
Since adverse events can easily be missed, it should be specified how formally they would have been recorded if they would have occurred: Was there a specific examination or questionnaire after the FM intervention?
Please specify how patient satisfaction was recorded (“pleased with the gains”) and correlated (“subjective reports appear to correlated with the statistically significant improvements in the subjective DHI scores and the objective metrics scores from the TUG”).
The most problematic limitation is stated as a conclusion (“Clinical improvements may be directly related to FM® intervention independently or in conjunction with a vestibular rehabilitation program or other unknown variables”). In fact, the setup (or at least the information provided in the manuscript) also allows the conclusion that the improvements were completely unrelated to FM.
The last paragraph should be less redundant (“Future research using rigorous methods and a control group must be conducted […] Future randomized clinical trial research using rigorous methods and study designs to include a control group and a comparison group must be conducted […]”).
Comments on the Quality of English Language
Few typos and unclarities, e.g. “functional physical” (comma missing), “Despite this limitation” (instead of these limiations) or “The only variable added to the participant”.
Author Response

(The authors gave the same response as above.)

Reviewer 3 Report
Comments and Suggestions for Authors
Comments to:
Short-Term Benefits from Manual Therapy as an Adjunct 2 Treatment for Persistent Postural-Perceptual Dizziness Symptoms (3PD): A Preliminary Prospective Case Series 4
I think it is commendable and interesting that we are trying to find more treatments as supplement to vestibular rehabilitation and provide a better and faster symptom improvement.
The study does not define what is short-term. It confuses. Is it right after the intervention? Or under treatment? After a month? Or 3 months which is the usual?
It says: previously receiving vestibular rehabilitation. Is that right? I perceive that this was done during ongoing vestibular treatment ? How long ago is it before the intervention and how long did the vestibular rehabilitation last? It is common for vestibular rehabilitation to last a minimum of 6 weeks and an assessment is made after 3 months. Was it like that here? When did this intervention begin?
As for the measuring instruments, DHI, TUG and TL measure DHI dizziness over the past month. When was it done in relation to the intervention? Are TUG and TL validated? Why not quality of life measurement?
I think the introduction with a long-running narrative about PPPD is too long and can be cut. However, it should state what previous studies have shown about the outcome of vestibular rehabilitation. Many studies show a dramatic improvement in DHI after 3 months of vestibular rehabilitation. In this context, the measured improvement in this study is not so remarkable. In any case, I'm starting to think that it might be not far off from a normal finding. The interesting thing about this intervention starts at line 99 where it justifies why fascia treatment might be interesting.
Line 110 reads: Current treatments already improve quality of life, allowing individuals to return to normal function within a shorter time frame [17]. Adding manual therapy to the current multi-modal treatment option may result in even faster progress. But the article has no time lines. It's a weakness. This study concludes that current multimodal treatment appears to reduce symptoms, there is no quality of life study.
Line 112:
The purpose of this case series was to investigate short-term outcomes of a manual therapy intervention focused at restoring mobility in the fascia using a pragmatically applied biomechanical approach, the Fascial Manipulation® (FM®), in subjects with PPPD.
Isn't this wrong? Shouldn't it have said:
The purpose of this case series was to investigate short-term outcomes of reducing dizziness symptoms using a manual therapy intervention focused at restoring mobility in the fascia using a pragmatically applied biomechanical approach, the Fascial Manipulation® (FM®), in subjects with PPPD.
Under 2.4. Plan of Care: Manual Therapy Intervention says: Patients were followed one to two times per week until symptoms were resolved or a maximum of eight visits. What symptom signals controlled the treatment? Were patients scored to evaluate when they finished? Were any patients treated for a week and some for eight weeks? Pain is described. Was it something they had or something they got from the treatment?
2.5. Plan of Care: Home Exercise Program I perceive that this describes ongoing vestibular rehabilitation and that this intervention did not occur afterwards, but during ongoing vestibular rehabilitation. Will it not be impossible to interpret what is due to what? Isn't this the opposite stated earlier where it says previously received vestibular treatment? Or is it both and?
4.1. The Effect of Manual Therapy on Each Metric in This Case Series 305
The dramatic significant improvement in DHI scores post-FM® (Table 1, Figure 2) 306 met the (MDC) of 17.18 [26]. Furthermore, participant DHI scores improved from pre- 307 scores of 53.8 ± 13, indicating high end “Moderate” dysfunction, to post-test scores of 10.2 308 ± 5.8, signifying lower end “Mild” dysfunction.[42]
I find this difficult to interpret when one does not know if pre is at some point during vestibular rehabilitation and post is undefined short-time line.
This section should be cut:The pathophysiology of PPPD remains unclear [50,51]. Theoretical approaches focus on visual, behavioral, and vestibular integration errors, which produce a reduced cortical integration of spatial cues in response to a triggering event. In general, symptoms associated with PPPD are worse in the upright position, exacerbated by changing visual stimuli, and may be impacted by active or passive head motions. People may complain of vague symptoms such as a full sensation in their head, haze, or cloudiness. During ambulation, they may feel unsteadiness or veer off course. It often takes simple contact to balance or stabilize themselves prior to movement [3,8,10].
Don't the following points belong during the introduction?
4.3. Value of Clinical Balance Assessments
A recent systematic review reported that center of pressure (CoP) tests might be more 366 sensitive when predicting falls than velocity measurements [32].
4.4. Prior Research Supporting the Soft Tissue Intervention
4.5. How Changes Might Have Incurred after FM®
393
4.6. Limiting other Potential Pain Generators for Continued Symptoms
In conclusion, the following is written about further study:
..if the FM® method, when added to current treatments, more quickly decreases dizziness, improves postural control, and enhances quality of life in those with persistent dizziness or PPPD symptoms.
I agree that this is an interesting study design, but I don't think this pilot answers whether there is a basis for doing a large randomized trial.
I think the authors have the data I miss. Combined with a different angle on improvement measured against other studies, a revised version should be interesting to publish.

Author Response
We thank you for your comments. We have provided responses to your comments in the attached file. We have provided the full paper where changes are identified using highlights for easier re-review. Furthermore, we have including Appendix A, B, and C. We hope that presenting the materials in this way, support and decrease your time as you re-review the manuscript. Thank you.

Round 2
Reviewer 2 Report
Comments and Suggestions for Authors
I find this revision unsatisfactory. If it was more balanced and conservative in its interpretation, the paper could be an interesting contribution to the ongoing discussion about PPPD treatment, but it remains one-sided and tendentious. The authors should be more transparent concerning the limited validity of this short-term uncontrolled case series.
A small change in this revision is “is considered” instead of “can be considered” in “PPPD (3PD) is considered a type of functional neurological disorder which may be correlated with significant cognitive-affective predisposing factors such as anxiety and depression”. This important sentence is still hidden in the limitations section instead of the introduction. The statement about functional gait abnormalities in the elderly does not at all refer to functional neurological disorders – this is a very different idea of “functional”. The authors still cite Popkirov et al but don’t really refer to its content, which describes the usually recommended treatment as active, psychological, and partly psychopharmacological.
It remains unclear, when the post FM assessments took place. Immediately after the interventions (which would carry a high risk for short term placebo effects, which is nowhere discussed)? After the HEP (which would make HEP an alternative explanation for the improvement but at least provide a longer follow-up)? Even without a control group, some kind of long-term assessment should be possible in 12 patients, and sustained effects would be a strong argument for a subsequent, more rigorous study.
Against this background, the authors should discuss the potential problems of passive interventions (see my first review) and explain how the FM method could overcome them.
Instead of discussing the limited validity of their data due to the inclusion of very old individuals with PPPD without properly documenting their comorbidities, the authors simply added the undisputed yet trivial statement, that “those with PPPD include a wide age distribution, suggesting PPPD may not be age dependent.”
Transparent limitations such as “Patient satisfaction and adverse events have not been documented systematically” are missing.
There are still several syntax errors, such as “functional physical” (there should be a comma, these are two DIFFERENT DHI subscales), “appear to correlated” or “the only variable added to the participant”.